# Analytical Modeling of Crack Widths and Cracking Loads in Structural RC Members

**David Z. Yankelevsky \*** , **Yuri S. Karinski and Vladimir R. Feldgun**

Faculty of Civil & Environmental Engineering, National Building Research Institute, Technion-Israel Institute of Technology, Haifa 32000, Israel; karinski@technion.ac.il (Y.S.K.); aefeldgo@technion.ac.il (V.R.F.)
**\*** Correspondence: davidyri@technion.ac.il

**Abstract:** Crack width is a major performance criterion in reinforced-concrete structures, in general, and is of utmost importance in ensuring bridge performance, in particular. A reliable theory-based method is required to assess crack widths and gain insight into their dependence on material, geometry, and loading parameters. A new, exact analytical method is proposed for a one-dimensional reinforced concrete element based on equilibrium, constitutive, and kinematic relationships, accounting for the geometrical and material behavior of the concrete and reinforcement. A linear interfacial bond stress slip is assumed to represents the small slips associated with the limited allowed crack width. Closed-form expressions have been developed and a wealth of information can be calculated immediately, such as the cracking load levels, the crack width dependence on the load level, the expected number of cracks, and the cracks spacing. The entire nonlinear force-displacement relationship of a cracked reinforced-concrete element may be depicted, demonstrating the tension-stiffening behavior that depends on the variations in the crack width throughout the loading history. Comparisons of the model with experimental data demonstrate very good agreement.

**Keywords:** tension-stiffening; cracking; bond–slip; linear constitutive relationships; nonlinear behavior; crack width

## 1. Introduction

### 1.1. Reinforced Concrete Cracking

Reinforced concrete (RC) is commonly used in civil engineering structures. As hardened concrete is relatively weak and brittle, it cracks easily under the action of a significant tensile stress. Steel rebars and/or prestressing steel are used to achieve the required tensile strength. The rebars also provide ductility to the structural element under tension and control cracking. When a slender concrete element with central longitudinal rebar is subjected to tensile loads at the rebar ends, the load is carried by both the concrete element and the rebar. Transfer of the tensile load from the rebar under tension to the surrounding concrete is enabled by the interfacial bond between the rebar and concrete; this phenomenon is enhanced when ribbed rebars are used. Increasing tensile load produces higher tensile stresses in concrete and causes it to crack. Variations in tensile stress along the element determine the location of new cracks and the corresponding cracking loads. This also affects the crack width and the structural member's axial stiffness, which decreases with an increasing number of cracks. At each cracking load level, stress redistribution occurs, and a new stress profile is formed along the concrete element and the rebar. Most studies have been conducted on RC members subjected to tension. However, flexural members may also be calculated as a tensile member with tensile reinforcement embedded in an equivalent concrete layer where its thickness depends on the concrete cover [1]. As cracking is an inherent characteristic of RC, crack width should be limited considering the functionality, durability, and appearance of the RC element. The maximum allowable crack width depends on the environmental conditions and the risk of corrosion attack [2]. Higher exposure risk to likelihood of carbonation, chlorides, freeze–thaw cycles, and chemical

attacks lead to smaller maximum allowable crack width. As a rule of thumb, this should not exceed 0.2–0.3 mm.

### 1.2. Cracking Problem—A Key Issue in Bridge Performance

The decks of most bridges, as well as many structural bridge components, are made of RC. For concrete bridges, reinforcing steel provides strength and ductility. The bond conditions affect the interaction, the tensile stress distribution, as well as the number of cracks and their widths. The fact that many bridges may be exposed to harsh climatic and environmental conditions bring about a need for stricter crack width requirements to assure adequate performance [3]. Freeze–thaw cycles add to these considerations, affecting bridge service life-cycle expectancy compared with regular buildings [2]. Therefore, to ensure the performance of bridge structures over time, corrosion risk should be avoided, and crack width should be calculated. Experimental studies based on the pullout of embedded corroded rebars [4,5] and on direct tension loading of corroded bars [1,6] verified that the effectiveness of bond is adversely affected by corrosion of the embedded steel rebar. This effect has been investigated using different analytical and numerical methods [7–9]. The local bond effect between steel rebars (corroded or non-corroded) and concrete is very important for the performance of structural elements of bridges in various ways. Of particular interest are cracks in the decks of continuous bridges over intermediate supports [3], as these open at the top of the deck, and therefore, the top reinforcement may be damaged, affecting the deck strength. Thus, crack width control is essential and should be included in the deck design [3,10,11]. In this case, the deck behaves almost as a pure tension tie in the longitudinal direction, assuming that the strain is constant over the depth of the slab [3]. This is even more evident in bridges with a concrete deck supported by steel girders. In many cases cracking problems can be reduced to a common representative model of tension members, as mentioned above [1,12,13].

### 1.3. Tension-Stiffening

Tension-stiffening is commonly related to the one-dimensional (1D) tensile behavior of a RC element that is composed of a concrete slender rod having a constant cross-section and central steel rebar along its longitudinal axis, where the rebar is subjected to equal tensile forces acting at its ends. Due to the bond between the rebar and the concrete, tensile stresses and strains develop in the axial direction of the concrete rod and the rebar, thus increasing the stiffness of the composite element in comparison with the stiffness of the bare rebar of the same length. This behavior is known as tension stiffening. The stress transfer mechanism between the rebar and the concrete due to interfacial bond stresses produces variable stresses and strains along the element in both the concrete rod and the rebar.

The 1D tension-stiffening model is an idealized representation of a fundamental problem in reinforced-concrete behavior. It considers the uniaxial tensile behavior of the concrete rod and the embedded steel rebar, as well as the interfacial bond–slip relationship. It follows the evolution of concrete discrete cracking and determines crack locations and widths. This information is required in the serviceability limit state to control maximum crack width criteria. It is also used in the analyses of the in-plane behavior of slabs, where the average tensile stress vs. strain relationship is based on the tension-stiffening mechanism.

The central cross-section of a RC element is a plane of symmetry. Although the end cross-sections of the concrete element are stress-free, and the entire load is carried by the rebar, other cross-sections along the RC element carry tensile stresses, both in the concrete and the rebar, depending on the geometrical and mechanical properties of the concrete and rebar and the interfacial bond properties. Due to the bond stresses between the rebar and the concrete, the tensile stress in the concrete is monotonically increasing from the element end planes towards the central cross-section at mid-length, whereas the axial stress in the rebar decreases accordingly. The highest magnitude of the tensile stress in the concrete is developed at the central cross section. The bond stress at a given coordinate along the rebar

depends on the interfacial relative displacement between the rebar and the concrete at that location, which is known as slip. The largest slip is developed at the end cross-sections.

A new crack is formed at the central cross-section of the RC element, when the tensile stress is equal to the tensile strength of concrete. Upon cracking, the RC element is split into two equal sub-elements, that are interconnected by the continuous bonded rebar. The width of the new formed crack is equal to the sum of rebar slips at both sides of the crack. The new boundary condition at the crack face is identical to the boundary condition at the end faces of the original RC element before cracking. Therefore, the variations of displacements, stresses, and strains along each sub-element are similar to those in the original element. Upon crack formation, a new topology of the RC element is formed that is characterized by two equal sub-elements with a crack in-between, and the stresses and strains along the element are redistributed accordingly. Upon cracking the stiffness of the element decreases as well. These changes define the initial conditions for the following loading. Upon further increase in loading, the stresses, strains, displacements, and crack width in the sub-elements increase as well. At a certain load level, the tensile stress in concrete, at the central cross-sections of the sub-elements, is equal to the tensile strength of concrete and another stage of cracking is developed at these cross-sections of each sub-element. At this stage, a new topology of four new sub-elements is formed. This process may continue further to the next cracking stages provided that the sub-element length enables increase in the tensile stress to the cracking strength level at the mid-section of each sub-element. This process is also limited by the maximum possible load level which produces the yield-limit stress $f_{sy}$ in the rebar.

Despite its simple geometry, this one-dimensional problem is rather complex and involves a degrading element stiffness at each cracking stage. The 1D tension-stiffening problem has gained much attention in numerous publications. The studies on the subject include analytical simplified approaches, as well as finite element and discrete element solutions e.g., [2,14–28]. A simple model for tension stiffening is described in the fib Model Code [2]. It focuses on the final state of full cracking, aiming at assessing the maximum crack width. it assumes a constant bond stress that is independent of the slip, although in a different chapter the Model Code presents the nonlinear bond stress–slip relationship that is based on experimental data. The constant bond stress assumption is the simplest representation of the concrete-rebar interaction and yields simplified linear variations of stresses along the concrete and rebar. Although it aims at providing simplified approximate expressions, it contradicts the experimental evidence indicating the variable bond stress with increasing slip.

The present paper aims at developing a simple analytical formulation that better represents the behavior than the constant bond–slip model. It expresses the entire behavior during the loading process and follows the loading and the cracking development, the variation of displacements, as well as stresses and strains and their dependence on the different cracking stages. Although the tension-stiffening behavior is strongly nonlinear, the present paper aims at demonstrating that major features of this behavior may be captured using linear constitutive relationships. The linear formulation may well represent part of the ascending branch of the bond–slip relationship and provide a simple closed form solution of the problem. This analysis clarifies the governing parameters and provides much insight into the nonlinear tension-stiffening behavior. Moreover, this model demonstrates that most of the nonlinear behavior inspected in experiments is not a result of the material nonlinearity but is derived from the variable topology of the RC element due to the increasing number of cracks at different loading levels. This nonlinear behavior is captured very well by the proposed formulation.

An exact finite-element (FE) model considering the rebar and its bond to the concrete had been developed by Yankelevsky [29], assuming that the surrounding concrete is relatively rigid with respect to the rebar. Later it has been extended to a two-phase, one-dimensional Tension-Stiffening Exact Finite Element (TSEFE) system that comprises the rebar, the surrounding concrete, and the bond interface. This is applied for uncracked and

cracked RC elements, using a linear or a piecewise-linear bond–slip relationship [30,31]. Using this exact FE model, an exact solution is obtained. The analysis requires a single or a limited number of finite elements along the RC element. Such analysis yields the nonlinear variation of displacements, stresses, and strains and follows the evolution of cracks. Comparisons of this model with numerous test data showed very good agreement. Although this analysis is considerably simpler and faster than common finite-element solutions, it is also based on numerical analysis, where the load is applied by small increments. It allows to follow the variation of stresses and displacements and focus on the formation of each crack, following its width variation. Using such step-by-step numerical procedures may introduce an accumulated error that may affect the precision of the final solution. Similar to all other numerical approaches, all the information regarding the RC element behavior, including the magnitude of loads at which cracks open, the increasing elongation and crack widths, etc., are revealed in the course of the analysis and cannot be predicted beforehand. Additionally, the effect of the different parameters cannot be assessed easily as it can be carried out in the case of a closed-form mathematical formulation.

The present paper is founded on the same principles of the exact stiffness approach and presents a new simplified analytical model for the same problem It aims at describing the entire nonlinear load-elongation processes during tension-stiffening by analytical linear expressions, thus avoiding the limitations of the existing approaches. Additionally, it can provide the entire load-elongation behavior right away.

## 2. New Approach

The new approach aims at formulating and analyzing the tension-stiffening behavior of a Uniaxial Reinforced Concrete Element (URCE) and at demonstrating the power of a linear formulation to describe the nonlinear non-monotonic behavior, as described above. The challenge for employing a simple closed form formulation for a detailed analysis that follows the nonlinear tension-stiffening behavior, leads to a linear elastic-based formulation both for the materials (steel and concrete) and for the bond–slip interfacial behavior. The rationale for this modelling is that the rebar may be modelled as linear elastic material until reaching its yield stress $f_{sy}$. The concrete is subjected to tension and its stress–strain behavior under tension may also be modelled very closely by a linear elastic model up to its cracking strength $f_t$, at which a crack is formed. The post peak tension softening in concrete is ignored for simplicity and upon reaching the tensile strength it is assumed that the crack fully opens. This simplification bypasses the intermediate stage of tensile microcracking development accompanied by stress decrease [32,33], which occurs over a very limited displacement. The continuous rebar crossing the crack connects the newly formed sub-elements and maintains a new state of equilibrium of all sub-elements.

A linear bond–slip relationship is assumed. This bond–slip relationship is appropriate for representing problems with limited slips. This is the case where a limited crack width is considered. The suitability and simplicity of the linear bond–slip relationship enable an analytical solution of the mathematical equations. Opposed to the constant bond stress assumption [2], the linear bond–slip relationship represents a good fit to the real nonlinear bond–slip relationship within the applicable slip range and may provide a realistic representation of the real behavior to a certain level of bond stresses and corresponding slips.

It should be noted that the complete nonlinear bond–slip relationship, which describes the entire pullout load–displacement behavior, is not required to represent the bond–slip relationship over a limited range of slips that characterizes the present tension-stiffening problem.

Due to symmetry, the slip at the central section of the URCE is zero and the largest slip is developed at the URCE ends. This maximum slip is rather small and is limited to a modest part of the ascending branch of the nonlinear bond–slip relationship.

Consider an advanced stage of loading, where cracks are developed along the URCE. The crack width is the sum of slips on both sides of the crack, and due to symmetry, the crack width is equal to twice the slip at the crack location. The width of this crack is limited

for serviceability considerations; Leonhardt [34] stated that the largest allowable crack width of structural elements shall not exceed 0.5 mm for mild ambient exposure conditions and 0.3 mm for more severe conditions. Recent provisions (e.g., the Model Code 2010 [2]) limit the allowable crack width of a structural RC element to 0.2–0.3 mm depending on the exposure conditions. It means that the maximum applicable slip magnitude is at the order of ~0.1–0.15 mm. This slip represents no more than 10–15% of the slip magnitude corresponding to the maximum bond stress, that is ~1 mm or even larger [35]. For that limited slip range, a linear bond–slip relationship is assumed.

The question is then: how can a linear-elastic formulation, using the linear representation of material properties and a linear bond–slip, represent the highly nonlinear tension-stiffening behavior? The clue is in the variable topology of the gradually cracking URCE that governs these global nonlinearities: according to the linear elastic formulation, any sub-element satisfies the linear constitutive relationships between cracking stages, but its behavior changes into another linear resulted behavior with different stiffness after a following cracking stage. The global nonlinear behavior results from the discrete jumps in the URCE extension due to the formation of new cracks, as well as due to the smaller stiffness of the shorter sub-elements that are formed after cracking. The formation of new sub-elements results in redistribution of the stresses and strains along the URCE. Thus, this overall nonlinear behavior is expressed in a piecewise linear behavior which represents the overall nonlinear behavior of the RC element. This will be described in detail in the following. The major advantage of this formulation is the holistic simple linear elastic formulation of a URCE (and similarly of a sub-element) that produces analytical expressions for the displacement, stresses, and strains at any point along the element and provides the solution components without performing any numerical computation. The analytical expressions resulting from this formulation demonstrate most clearly the effects of the governing parameters on the tension-stiffening behavior.

### 3. Proposed Model

#### 3.1. Model Description

Consider a URCE of length 2L [mm] (Figure 1). The concrete cylinder diameter is $D_c$ [mm] and its cross-section area is $A_c$ [mm$^2$]; the rebar diameter is $D_s$ [mm] and its cross-section area is $A_s$ [mm$^2$]. The concrete and steel Young's moduli are $E_c$ and $E_s$ [MPa], respectively. The rebar-concrete interface satisfies a linear bond-stress relationship. Tensile forces P [N] are applied at both ends of the rebar, where the concrete end faces are stress free. At a certain level of loading, a crack opens at the URCE central cross-section. The URCE is subdivided into two equal sub-elements which are similar to the original URCE. The model formulation referring to the entire URCE of length 2L prior to any cracking also refers to each of the two sub-elements of length L. In general, the basic formulation of a URCE of length 2L refers to the following formed sub-elements of length $2\ell$ (where $\ell$ = L, L/2, L/4, L/8, etc.). Each sub-element is characterized by stress free end faces of the concrete, both at the URCE ends and at the stress-free faces of the concrete at a crack between the sub-elements. The principles of the URCE and sub-elements formulation are outlined in the following section.

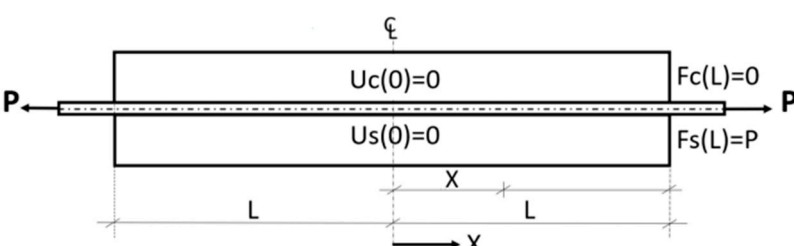

**Figure 1.** The URCE.

### 3.2. Mathematical Derivation of the Basic Model

### 3.2.1. Definitions and Derivation of the Governing Equation

Consider a URCE of length 2L as shown in Figure 1. Denote the local displacements in the rebar and concrete as $U_s(x)$ and $U_c(x)$, respectively, and define the local slip $S(x)$ as:

$$S(x) = U_s(x) - U_c(x) \tag{1}$$

Due to symmetry, half of the URCE ($0 \leq x \leq L$) is considered. The origin of the coordinate system is set at the central cross-section. Figure 1 specifies the boundary conditions. The bond–slip relationship satisfies Equation (2).

$$\tau(x) = AS(x). \tag{2}$$

where $\tau(x)$ is the local bond stress [MPa], $S(x)$ is the local slip [mm] and A [MPa/mm] is the slope of the bond–slip relationship.

The axial strains in the concrete and in the rebar at distance x are, respectively:

$$\varepsilon_c(x) = \frac{dU_c(x)}{dx} \tag{3}$$

$$\varepsilon_s(x) = \frac{dU_s(x)}{dx} \tag{4}$$

The differential equilibrium equations for a concrete and a rebar differential elements of length dx, located at distance x along the element, are:

$$\frac{dF_c(x)}{dx} = -\pi D_s \tau(x) \tag{5}$$

$$\frac{dF_s(x)}{dx} = \pi D_s \tau(x) \tag{6}$$

The axial forces in the concrete and steel at a distance x are, respectively:

$$F_c(x) = E_c A_c \varepsilon_c(x) \tag{7}$$

$$F_s(x) = E_s A_s \varepsilon_s(x) \tag{8}$$

Substituting Equations (3), (4), (7) and (8) into Equations (5) and (6), yields:

$$\frac{d^2 U_c(x)}{dx^2} = -\frac{\pi D_s}{E_c A_c} \tau(x) \tag{9}$$

$$\frac{d^2 U_s(x)}{dx^2} = \frac{\pi D_s}{E_s A_s} \tau(x) \tag{10}$$

Substituting Equations (9) and (10) into the second derivative of Equation (1) yields:

$$\frac{d^2 S(x)}{dx^2} = \frac{\pi D_s [1 + n\rho]}{E_s A_s} \tau(x) \tag{11}$$

where:

$$n = \frac{E_s}{E_c} \tag{12}$$

$$\rho = \frac{A_s}{A_c} \tag{13}$$

Equation (11) is a differential equation with two unknown functions $S(x)$ and $\tau(x)$, that are related by the bond–slip relationship (Equation (2)).

### 3.2.2. Solution of the Governing Equation

Substituting the bond–slip relationship (Equation (2)) into Equation (11) yields the following second order ordinary differential equation:

$$\frac{d^2 S(x)}{dx^2} = \alpha^2 S(x) \tag{14}$$

where $\alpha$ [1/mm] is the characteristic length:

$$\alpha^2 = \frac{\pi D_s(1 + n\rho)}{E_s A_s} A \tag{15}$$

The solution of Equation (14) is:

$$S(x) = D_1 e^{\alpha x} + D_2 e^{-\alpha x} \tag{16}$$

From the boundary condition S(0) = 0 we find that $D_1 = -D_2 = D$, and Equation (16) reads:

$$S(x) = 2D \sin h(\alpha x) \tag{17}$$

Equations (9) and (10) may be rewritten for this case:

$$\frac{d^2 U_c(x)}{dx^2} = -\frac{\pi D_s}{E_c A_c} A S(x) \tag{18}$$

$$\frac{d^2 U_s(x)}{dx^2} = \frac{\pi D_s}{E_s A_s} A S(x) \tag{19}$$

Using Equation (17), the solution of Equations (18) and (19) becomes:

$$U_c(x) = -\frac{2Dn\rho}{1 + n\rho} \sinh(\alpha x) + F_1 x + F_2 \tag{20}$$

$$U_s(x) = \frac{2D}{1 + n\rho} \sinh(\alpha x) + G_1 x + G_2 \tag{21}$$

Substituting Equations (20) and (21) into Equation (1) yields:

$$S(x) = 2D\sinh(\alpha x) + (G_1 - F_1)x + (G_2 - F_2) \tag{22}$$

Comparing Equation (22) with Equation (17), and noting that Equation (22) is valid for any x yields:

$$G_1 = F_1; G_2 = F_2 \tag{23}$$

The boundary conditions are:

a.   Symmetry at x = 0:

$$U_s(0) = 0 \tag{24}$$

This leads to:

$$G_2 = F_2 = 0 \tag{25}$$

Note that the condition $U_c = 0$ is obtained automatically from Equations (1) and (24).

b.   Rebar loading at x = L:

$$\frac{dU_s(L)}{dx} = \frac{P}{E_s A_s} \tag{26}$$

It yields:

$$\frac{2D\alpha}{1+n\rho}\cosh(\alpha L) + G_1 = \frac{P}{E_s A_s} \tag{27}$$

c.   Free concrete surface at x = L:

$$\frac{dU_c(L)}{dx} = 0 \tag{28}$$

It yields:

$$F_1 = \alpha \frac{2Dn\rho}{1+n\rho}\cosh(\alpha L) \tag{29}$$

$G_1$, $F_1$, and D may be determined from Equations (27) and (29):

$$G_1 = F_1 = \frac{n\rho}{1+n\rho}\frac{P}{E_s A_s} \tag{30}$$

$$D = \frac{P}{E_s A_s}\frac{1}{2\alpha\cosh(\alpha L)} \tag{31}$$

These constants are used to obtain explicit expressions for the variation of slip S(x), displacements in concrete $U_c(x)$ and rebar $U_s(x)$, axial stress in concrete $\sigma_c(x)$, stress in rebar $\sigma_s(x)$, and bond stress $\tau(x)$:

$$S(x) = \frac{P}{E_s A_s}\frac{\sinh(\alpha x)}{\alpha\cosh(\alpha L)} \tag{32}$$

$$U_c(x) = -\frac{n\rho}{1+n\rho}\frac{P}{E_s A_s}\left(\frac{\sinh(\alpha x)}{\alpha\cosh(\alpha L)} - x\right) \tag{33}$$

$$U_s(x) = \frac{1}{1+n\rho}\frac{P}{E_s A_s}\left(\frac{\sinh(\alpha x)}{\alpha\cosh(\alpha L)} + n\rho x\right) \tag{34}$$

$$\sigma_c(x) = \frac{\rho}{1+n\rho}\frac{P}{A_s}\left(1 - \frac{\cosh(\alpha x)}{\cosh(\alpha L)}\right) \tag{35}$$

$$\sigma_s(x) = \frac{1}{1+n\rho}\frac{P}{A_s}\left(\frac{\cosh(\alpha x)}{\cosh(\alpha L)} + n\rho\right) \tag{36}$$

$$\tau(x) = \frac{PA}{E_s A_s}\frac{\sin h(\alpha x)}{\alpha\cosh(\alpha L)} \tag{37}$$

### 3.3. Features of the Proposed Model

Consider a sub-element of length $2\ell$; it represents the URCE of length 2L prior to cracking, or a typical sub-element of length $2\ell$, where $2\ell = L$ for this URCE with a single central crack, or $2\ell = L/2$ for the case of a URCE with 3 equally spaced cracks, etc. In this section we examine the variation of the major parameters along $\ell$, investigate the governing parameters and describe a procedure for an efficient depiction of the force–elongation diagram.

#### 3.3.1. Variation of Major Parameters along the URCE

The present model yields expressions for an easy and straightforward evaluation of the effect of different parameters. It also provides an immediate view of the variations of these parameters along the URCE. Insight is gained on the bond stress transmission and its effect on the other parameters. This is demonstrated herein in a study of two parameters: the variation of slip and of the tensile stress in concrete along the URCE. The analysis refers to a URCE of length 2L, however the same results are applicable to any stage of loading referring to a sub-element of length $2\ell$.

a.   Slip function (Equation (32))

The maximum slip develops at x = L and its value is:

$$S_{max} = \frac{P}{E_s A_s} \frac{\tan h(\alpha L)}{\alpha} \tag{38}$$

Examination of this expression shows that the non-dimensional relationship (Equation (39)) approaches a constant value of 1 very rapidly, where its value is equal to 0.99 at $\alpha L = 2.6$.

$$\frac{\alpha S_{max} E_s A_s}{P} = \tanh(\alpha L) \tag{39}$$

This means that for a long URCE, the maximum slip depends on the applied load but is independent of the URCE length.

Figure 2 shows the variation of the normalized slip along a URCE of length 2L, for a typical characteristic length ($\alpha = 0.02$) and different values of L:

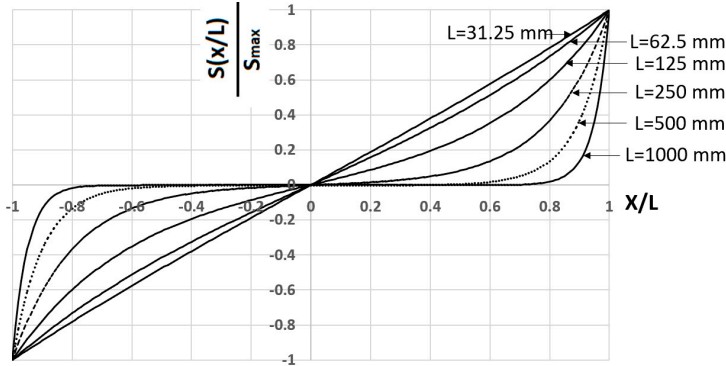

**Figure 2.** Variation of the slip along the URCE.

Figure 2 shows that for a long URCE, slip is not developed along a considerable part of the URCE. Slips are developed along a certain distance from the element ends that is shorter than L; this is the transfer length of stresses between the rebar and the concrete. For shorter elements, the curvature of the slip variation function is more moderate and for rather short elements it approaches a linear variation along the URCE.

b.    Tensile stress in concrete (Equation (35))

At the URCE boundaries, the tensile stress in concrete is zero and its magnitude increases towards the center of the URCE (x = 0), where the tensile stress reaches its maximum magnitude. When this maximum stress becomes equal to the tensile strength of concrete $f_t$, a crack is formed, and the element is split into two sub-elements as described above. Figure 3 depicts the normalized tensile stress along the URCE axis, for different URCE half-lengths L with a typical characteristic length ($\alpha = 0.02$). This shows that the growth of tensile stress depends on the URCE length; for shorter URCEs the stress gradually increases towards the center at a decreasing slope and reaches its maximum value at the URCE center. For longer URCEs the stress reaches the maximum magnitude at a certain distance that is shorter than the URCE half-length and depends on L. Hence, the maximum stress magnitude exists along a significant central part of the URCE. This means that although in shorter elements the highest tensile stress is clearly developed at the central cross-section and the next crack will open at that location, in longer elements there is a zone of constant stress where the next crack may open. The relationship between the variation of slip (Figure 2) and the variation of the tensile stress in concrete (Figure 3) should be mentioned: the slip variation in the present model is proportional to the bond stress variation, hence the rate of change in the bond stress is tightly connected to the rate of change of the tensile stress in concrete, and when the slip and corresponding bond stress drop to zero (Figure 2), the tensile stress reaches its maximum magnitude (Figure 3).

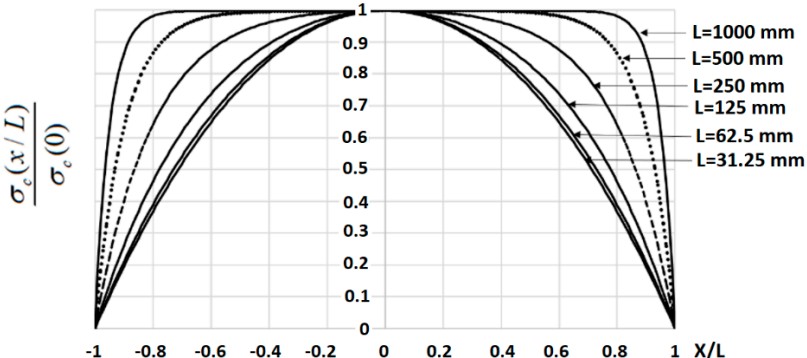

**Figure 3.** Variation of the tensile stress in concrete along the URCE.

### 3.3.2. Investigation of Governing Parameters

In this section several parameters, which affect the analysis results are discussed, including the nonlinear bond–slip relationship, the validity of the linear bond–slip assumption in the present model, the bond–slip relationship related to tension-stiffening experimental program, and the characteristic length $\alpha$.

a.   Nonlinear bond–slip relationship:

The bond–slip relationship is commonly determined in pullout tests where a rebar that is centrally embedded in a concrete cylinder is pulled out and the total force at any stage of loading and the corresponding slip are measured. These tests provide a bond-stress vs. slip relationship that is characterized by an ascending nonlinear curve up to the maximum bond-stress, developed at a slip of 1–2 mm, and followed by a gradual descending bond-stress vs. slip relationship at larger slips. The experimental data have been compiled into an empirical expression of a power law for the ascending branch and a linear expression for the descending branch [2]. Comparison of numerous experimental results with the proposed empirical expression indicates some variations between the different curves, due to different parameters (e.g., concrete strength, rebar diameter and shape, length of bonded area), test setup, etc. To enable analysis and comparisons with different experimental results, it is useful to present the results in a non-dimensional coordinate system $\tau(x)/\tau_{max}$ vs. $S(x)/S_{max}$, where $\tau_{max}$ is the maximum bond stress and $S_{max}$ is the corresponding slip. Figure 4 depicts four curves from three different references (Viwathanatepa et al. [36], Soroushian et al. [37] and Leibovich et al. [38]). The nonlinear non-dimensional ascending curves connect the origin with the point of maximum normalized stress. Even within the non-dimensional description, some deviation between the curves is observed, especially along the intermediate range of the ascending branch.

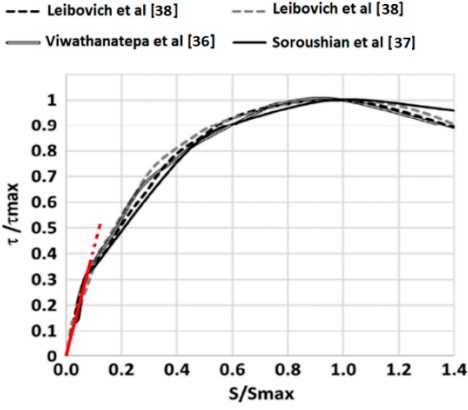

**Figure 4.** The linear bond–slip relationship at limited slips.

b.   Validity of a linear bond–slip model for the present tension stiffening problem:

A linear bond–slip relationship may represent the early part of the nonlinear experimental bond–slip relationship. The red line in Figure 4 shows that the linear relationship may well replace the nonlinear curve within limited slips up to S/Smax = ~0.1 that means a slip of 0.1 mm where $S_{max}$ = 1 mm or 0.2 mm where $S_{max}$ = 2 mm. Such a linear representation cannot be extended to the entire slip range of the pullout process; however, it may represent a major part of the tension-stiffening loading program as discussed below.

A slip of 0.1 mm is related to a crack width of 0.2 mm. That crack width is at the upper limit of the allowable crack width according to the serviceability limit state (see Section 2). Therefore, this linear representation may suit the slip range in the present problem. In a case where larger cracks are considered, an extended bond–slip relationship is preferrable.

It is important to note that although the bond–slip relationship plays a key role in the tension-stiffening problem, bond–slip tests have not been carried out in conjunction with tension-stiffening experimental programs, thus these parameters should be assumed based on available parameters (e.g., concrete uniaxial compressive strength, rebar diameter, concrete cover, etc.). Regarding the linear bond–slip curve, a single parameter is required, that is the slope A [MPa/mm]. The power law representing the ascending branch in the fib Model Code [2] cannot provide this information as its initial slope is ∞. The slope may be assumed from depicting an approximate linear relationship representing the early relevant stress–slip range. Attempt to determine the slope A indicates that it may range between 65–200 MPa/mm. Examination of numerous experimental bond–slip curves [38] yields a similar range (25–200 MPa/mm). This means that the slope A may vary quite largely, and its value may affect the calculated results; this will be further examined in the following section. It should be noted that a higher value of A corresponds to smaller slips.

c.    Characteristic length $\alpha$:

The characteristic length $\alpha$ (Equation (15)) depends on the rebar and concrete cross-section areas, concrete and steel Young's moduli, and slope of the bond slip curve. The range of the characteristic length parameter is evaluated for a typical case with the following data: $\rho$ = 1%, n = 7, $E_s$ = 210,000 MPa, the slope A varies within the range 50–200 MPa/mm, and the rebar diameter varies between 8–20 mm. The results in Table 1 indicate that $\alpha$ varies roughly within 0.01–0.03 [1/mm].

**Table 1.** Variation of the characteristic length $\alpha$ [1/mm].

| A [MPa/mm] | 50 | 100 | 150 | 200 |
|---|---|---|---|---|
| $D_s$ [mm] | | | | |
| 8 | 0.011 | 0.016 | 0.019 | 0.027 |
| 10 | 0.010 | 0.014 | 0.017 | 0.020 |
| 12 | 0.009 | 0.013 | 0.016 | 0.018 |
| 14 | 0.008 | 0.012 | 0.015 | 0.017 |
| 16 | 0.008 | 0.011 | 0.014 | 0.016 |
| 18 | 0.007 | 0.010 | 0.013 | 0.015 |
| 20 | 0.007 | 0.010 | 0.012 | 0.014 |

### 3.3.3. Major Characteristics of The tension-Stiffening Problem

From the model formulation (Section 3.2.2) general expressions of several major parameters are derived in the following. The parameters may shed light on the URCE behavior.

a.    Shortest distance between cracks:

It has been shown that upon increasing the applied load, the tensile stress in concrete increases, until its magnitude at the URCE mid-length is equal to the concrete tensile

strength $f_t$, where a new crack is formed. The new crack splits the URCE into two equal sub-elements of half-length L each, and upon further loading, the tensile stress at mid-length of each of these sub-elements increases as well until a following cracking stage is formed. A key parameter in the tension-stiffening behavior is the shortest length between cracks, which determines how many cracking stages occur.

Define the minimum half length of the sub-element as $L_{min}$ and assume that this sub-element will be cracked at its center upon reaching the maximum load magnitude. Thus, the shortest distance between cracks becomes $L_{min}$. This length is sufficiently short, such that the maximum developed tensile stress at its mid-length is still reaching the concrete cracking strength at the stage where the rebar is almost reaching its yield stress. Using Equation (35) and referring to the stress at the sub-element mid-length that is equal to $f_t$, the expression for $L_{min}$ [mm] is obtained:

$$L_{min} = \frac{1}{\alpha} \cos h^{-1} \left( \frac{1}{1 - \frac{f_t}{f_{sy}} \frac{1+n\rho}{\rho}} \right) \tag{40}$$

$L_{min}$ may be determined a priori and provide important information regarding the number of expected cracks and/or the number of sub-elements formed after cracking. To obtain that information, one should simply round down the nearest integer of $L/L_{min}$ and compare it with the series representing the possible number of sub-elements that are developed during the tension-stiffening process of half-length of the URCE (i.e., 2, 4, 8, 16, 32). For example, consider a URCE with the following data: length 2L = 762 mm, rebar diameter 10 mm, concrete diameter 93 mm, Young's moduli of the concrete and steel are, respectively 27,794 MPa and 158,970 MPa, and the bond–slip slope A = 174 MPa/mm. From Equation (18) we find that $L_{min}$ = 81.68 mm, hence $L/L_{min}$ = 4.66. This indicates that the maximum number of sub-elements of half-length of the URCE is 4; at its fully cracked state, the URCE will include eight sub-elements of equal length that are split by seven cracks. That cracking topology is reached after three cracking levels: the first cracking level where a single central crack is developed, which split the URCE of length 2L into two sub-elements of length L; the second cracking level where two more cracks are developed at the centers of the two sub-elements, thus forming four sub-elements of length L/2, and the third cracking level where four more cracks are formed at the centers of the four sub-elements, thus finally forming eight sub-elements, separated by seven cracks.

b.    Cracking Load $P_{cr}$:

Extracting the force P from Equation (35) for sub-element half-length of L, L/2, L/4, etc. at the stage where $\sigma_c(0)$ = $f_t$, yields the expression for the cracking load $P_{cr}$ for the different cracking stages:

$$P_{cr} = \varepsilon_t (E_c A_c + E_s A_s) \frac{\cosh(\alpha\ell)}{\cosh(\alpha\ell) - 1}, \ell = L, \frac{L}{2}, \frac{L}{4}, \frac{L}{8}, \text{etc.} \tag{41}$$

where $\varepsilon_t$ is the concrete cracking strain.

Equation (41) shows that the cracking load $P_{cr}$ is composed of a constant term $\varepsilon_t(E_c A_c + E_s A_s)$ and a variable term $\frac{\cosh(\alpha\ell)}{\cosh(\alpha\ell)-1}$ which depends on the constant characteristic length $\alpha$ and the sub-element half-length $\ell$. The variable term is larger than unity and may be interpreted as an amplification factor with larger effect at shorter values of $\alpha\ell$ (Figure 5).

This means that for a slender URCE of length 2L, the second cracking load calculated for half-length L/2 will not differ much from the first cracking load that is calculated for half-length L, and at shorter sub-elements the amplification parameter will determine a higher cracking load. This can be demonstrated using the same data given in Section 3.3.3-a above. Table 2 shows the calculated cracking loads $P_{cri}$ for the three cracking levels calculated above. The calculated amplification factors (Figure 6) are given in the fourth column and

show that there is a difference of less than 3.5% between $P_{cr2}$ and $P_{cr1}$, but a difference of almost 34% between $P_{cr3}$ and $P_{cr1}$.

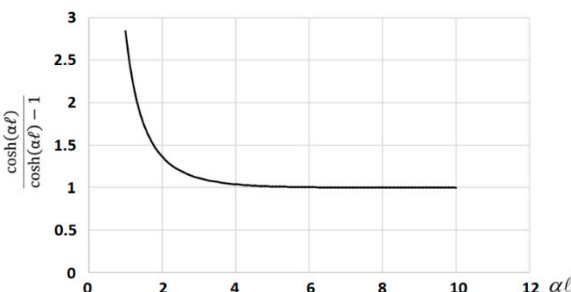

**Figure 5.** Amplification factor.

**Table 2.** Cracking loads.

| Cracking Load Level i | | $\ell$ | $\alpha\ell$ | $P_{cri}$ (kN) |
|---|---|---|---|---|
| (1) | (2) | (3) | (4) | (5) |
| 1 | L | 381 | 1.0005 | 18.78 |
| 2 | L/2 | 190.5 | 1.0337 | 19.40 |
| 3 | L/4 | 95.25 | 1.3357 | 25.07 |

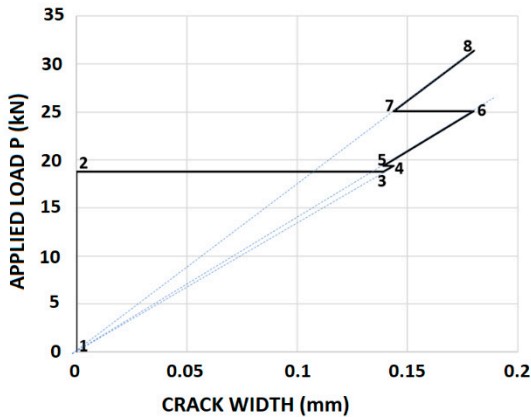

**Figure 6.** Variation of crack width.

c.    Crack widths at the cracking load levels $P_{cri}$:

The crack width is another important parameter that can be evaluated right away. It may be calculated using Equation (32), assuming that the crack width is equal to the sum of slips from both crack faces, and since these slips are identical, it turns out that the crack width, $\Delta$, is:

$$\Delta = \frac{2P_{cr}}{E_s A_s} \frac{\tan h(\alpha\ell)}{\alpha} \tag{42}$$

where $P_{cr}$ is the cracking load at a certain cracking stage and $\ell$ is half length of the sub-element at this cracking stage.

Table 3 calculates the widths of cracks at the different cracking stages. This indicates that in the present example, the crack width up to $P_{cr2}$ and beyond is about 0.14 mm. The corresponding slip is therefore 0.07 mm. That limited slip justifies the linear bond–slip assumption and good results are expected using this simple model.

**Table 3.** Cracking stages and crack widths.

| Stage | Crack Width $\Delta_1$ | Length $\ell$ | Scheme |
|:---:|:---:|:---:|:---:|
| 1 | $\Delta_1 = 0$ | | |
| 2 | $\Delta_2 = 0$ | $\ell = L$ | |
| 3 | $\Delta_3 = \frac{2P_{cr1}}{E_s A_s} \cdot \frac{\tanh(\alpha\ell)}{\alpha}; \ell = L/2$ | $\ell = L/2$ | |
| 4 | $\Delta_4 = \frac{2P_{cr2}}{E_s A_s} \cdot \frac{\tanh(\alpha\ell)}{\alpha}; \ell = L/2$ | $\ell = L/2$ | |
| 5 | $\Delta_5 = \frac{2P_{cr2}}{E_s A_s} \cdot \frac{\tanh(\alpha\ell)}{\alpha}; \ell = L/4$ | $\ell = L/4$ | |
| 6 | $\Delta_6 = \frac{2P_{cr3}}{E_s A_s} \cdot \frac{\tanh(\alpha\ell)}{\alpha}; \ell = L/4$ | $\ell = L/4$ | |
| 7 | $\Delta_7 = \frac{2P_{cr3}}{E_s A_s} \cdot \frac{\tanh(\alpha\ell)}{\alpha}; \ell = L/8$ | $\ell = L/8$ | |
| 8 | $\Delta_8 = \frac{2P_{sy}}{E_s A_s} \cdot \frac{\tanh(\alpha\ell)}{\alpha}; \ell = L/8$ | $\ell = L/8$ | |

Figure 6 presents a general scheme describing the evolution of cracks and their width change in a case of three cracking load levels that form seven cracks before rebar yielding. The scheme may be extended for more cracking stages if applicable. The appropriate parameters of the cracking load $P_{cri}$ (i is the cracking stage) and $\ell$ is presented for each stage.

Figure 6 shows the crack width variation with increasing loading, up to the yield load $P_{sy}$ of the steel rebar ($P_{sy} = A_s \cdot f_{sy}$).

Figure 6 shows the non-monotonic variation of crack width with increasing load. The piecewise linear curve indicates the different loading stages that are presented in Table 3: at loads lower than $P_{cr1}$ there is no crack (line 1–2 in Figure 6). A first crack opens at $P_{cr1}$ (line 2–3) and further loading increases the crack width (line 3–4).

At $P_{cr2}$ more cracks open and the crack width decreases (line 4–5). Further loading up to $P_{cr3}$ increases the crack width (line 5–6), and upon reaching $P_{cr3}$ three more cracks open and a pronounced decrease in crack width is observed (line 6–7).

Further loading up to the yield stress level of the rebar does not open any more cracks and the width of the cracks increase (line 7–8). Note that crack width with increasing load between cracking stages (lines 3–4, 5–6, 7–8) is not growing at the same rate: at higher cracking stages, the rate of crack increase with loading is smaller.

The dotted lines in Figure 6 show the rate of crack width increase and demonstrate that these straight lines emerge from the coordinates origin, as expected from the linear based formulation of the model. Despite the complex variation of the crack width with the increasing load, it is important to note that as the number of cracks increases at the different cracking stages, and the width of a crack may decrease and then increase as shown in Figure 6, the total crack width increases at higher cracking stages (Figure 7).

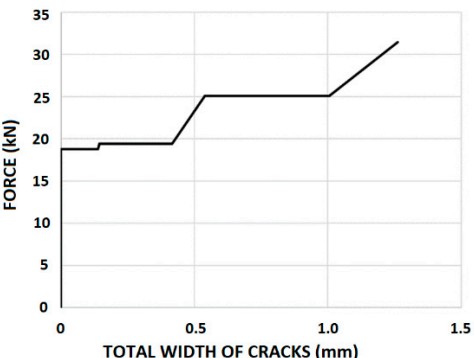

**Figure 7.** Total cracks width.

It is interesting to note that the proposed model using the linear constitutive relationships and the above formulation can follow the non-monotonic behavior of crack width opening and partial closure during loading. It should be mentioned that this behavior has not been identified so far, except for a single experiment [39] mainly because it requires careful measurements, which place the sufficiently sensitive gauges mounted at the location of a crack that has not been opened yet. To overcome that difficulty, Wollrab et al. [39] prepared a notch to precisely control the crack opening location and installed gauges bridging over that location thus enabling the measurement of the crack width [31].

### 3.3.4. The Force–Elongation Relationship

In most of the tension-stiffening tests, the applied force and the resulting elongation at the rebar ends are measured and the force–elongation relationship is depicted.

The proposed model can easily produce the force–elongation relationship. Due to the linear force–elongation relationship for a given cracking topology, it is sufficient to calculate the elongation at the cracking load levels, prior and after the new crack(s) opening. Following the stages shown in Table 3 and using Equation (34), the following expression may be used to calculate the elongation of a sub-element of length $\ell$:

$$\Delta\ell = 2U_s(\ell) = \frac{2}{1+n\rho} \cdot \frac{P\ell}{E_s A_s}\left(\frac{\tanh(\alpha_1\ell)}{\alpha_1\ell} + n\rho\right) \tag{43}$$

The total URCE elongation is the sum of $\Delta\ell$ on all sub-elements.

Figure 8 shows the force–elongation relationship for the given set of data (solid line), and its comparison with two experiments, no. 2 and no. 7, with no transverse reinforcement (shown in red circles and blue triangles) that had been carried out by Rizkalla et al. [40]. The dimensions of both specimens were 178/305/762 mm, and they were reinforced with eight 11.3 mm diameter rebars. The material properties are: $E_C$ = 27,794 MPa; fc = 34.5 MPa; ft = 2.62 MPa; $E_S$ = 199,955 MPa.

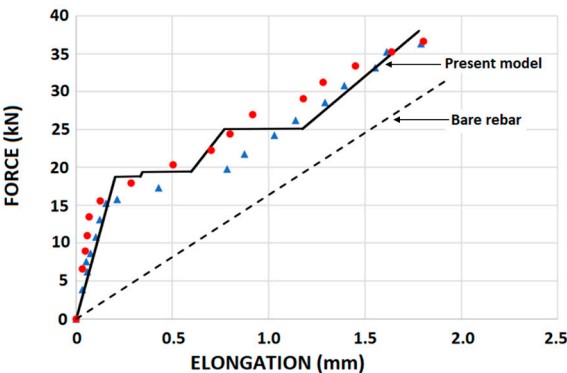

**Figure 8.** Comparison of Rizkalla et al. tests [40] with model analysis.

The force–elongation relationship of the bare rebar is shown in a dashed line for comparison. This comparison shows very good agreement between the present model and the experimental results. During the experiments elongation measurements were taken at discrete loading stages, thus the elongation's abrupt increase upon cracking is not described. Nevertheless, the calculated curve is in very good agreement with the general trend of the experimental measured data.

## 4. Comparisons with Test Data and Analytical Results

### 4.1. Comparison with Computer Software

Concrete elongation has been rarely measured in tension-stiffening testing of RC elements, although it has been commonly measured in concrete panels subjected to in-plane axial and shear loading by using LVDT's attached to fixed points in the concrete panel or using clip gauges e.g., [41]. ATENA [42], a computer software for nonlinear analysis of concrete structures, was used to calculate tension-stiffening tests that had been performed by Hartl [41]. Hartl's tests were carried out on a 750 mm long concrete prism with a cross section of 80/80 mm, with a single 12 mm central rebar. The elastic moduli of the concrete and the steel rebar are 29,000 MPa and 210,000 MPa, respectively. The concrete tensile strength and uniaxial compressive strengths are 3.1 MPa and 22.95 MPa, respectively. The rebar yield stress is 460 MPa [31]. The LVDT's were installed on the concrete outer face, measuring the concrete elongation over a central span of 500 mm (Figure 9). LVDT clips were installed 125 mm from the concrete specimen ends.

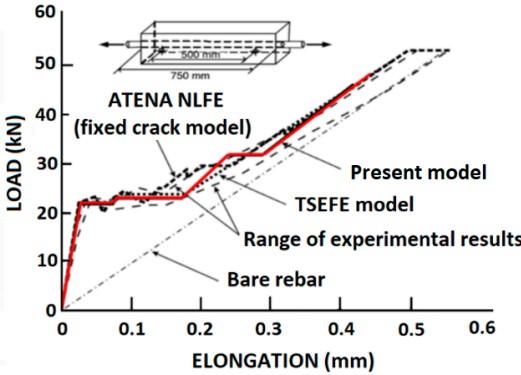

**Figure 9.** Comparison of analysis methods with Hartl's test results.

Figure 9 shows that the present model analysis falls nicely within the bounds of the measured data and shows an even somewhat better correspondence than obtained by the nonlinear analysis using ATENA.

### 4.2. Concrete Elongation—Comparison with Analytical Methods and Empirical Expressions

The comparison of the same test data obtained by Hartl [41] with the present model, is extended to comparisons with different analytical methods, including the fib Model Code [2], the modified compression field theory (MCFT) [43], and the Belarbi and Hsu model [44].

Figure 10 shows that the present model is in good agreement with the test data and with the results of the analytical and empirical expressions. This example demonstrates the model ability to predict the concrete elongation during loading.

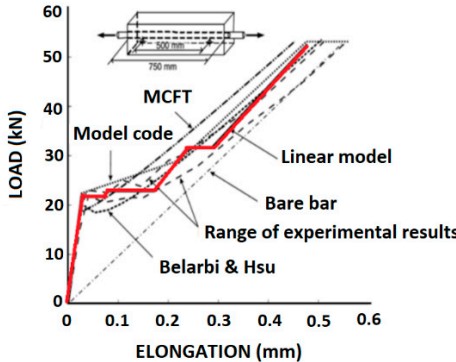

**Figure 10.** Comparison of Hartl's test results with different models.

*4.3. Comparisons with Tests by Houde and Mirza*

Houde and Mirza performed tension-stiffening tests on an 1829 mm long cylindrical specimen (data is reported by Chan et al. [45]) with the following data: $E_C$ = 24,823 MPa; ft = 1.38 MPa; $E_S$ = 200,000 MPa; $D_S$ = 25.4 mm; $D_C$ = 152.4 mm, and on an 838 mm long prismatic specimen with the following data: $E_C$ = 23,788 MPa; ft = 2.12 MPa; $E_S$ = 200,000 MPa. Figures 11 and 12 compare the tests results with the present model analysis and good correspondence is obtained.

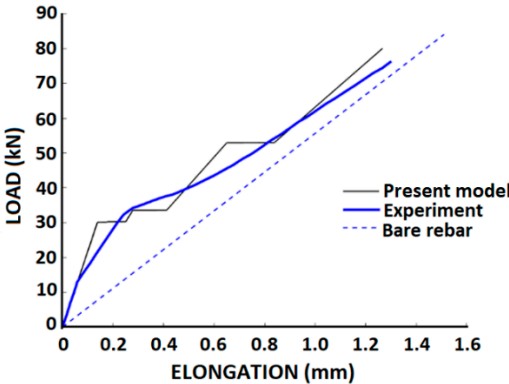

**Figure 11.** Comparison with Houde and Mirza: axi-symmetrical URCE.

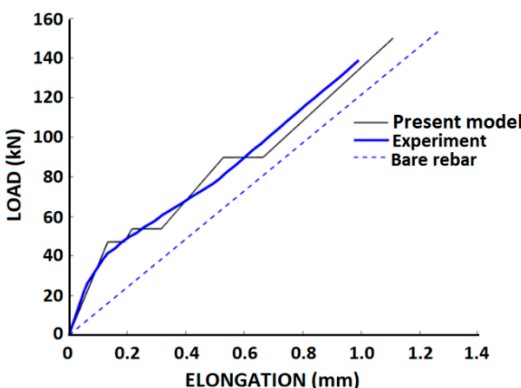

**Figure 12.** Comparison with Houde and Mirza: prismatic URCE.

## 5. Summary and Conclusions

The one-dimensional tension-stiffening problem of a slender RC element, composed of a concrete cylinder with a centrally embedded longitudinal rebar, is investigated. The rebar ends are subjected to equal tensile forces in a pull-pull mode. The tensile behavior of concrete and the steel rebar may be well represented by linear stress–strain models. In this paper we argue that although the bond–slip relationship in its general form is nonlinear, it

may be approximated by a linear model to follow the small slips range up to about 0.1 mm. This slip range is sufficient to represent the cracking in the present problem during loading and follow the opening and extension of cracks with widths that are up to 0.2 mm. Such crack width is in accordance with the crack width design at the serviceability limit state of reinforced concrete bridges.

The linear bond–slip relationship enables to formulate a simple closed form solution of the problem considerably closer to reality than the commonly assumed constant bond stress that is independent of the slip. The solution provides insight on the effect of key parameters on the RC element behavior, as well as obtain a priori basic information on the RC element, such as the number of expected cracks and their widths, the cracking loads, and the load-elongation relationship that is commonly measured in tension-stiffening tests. Comparison of the model with test results as well as with nonlinear finite-element analyses and common analysis methods show good agreement and demonstrate the ability of the present model to follow the nonlinear behavior exhibited in a tension-stiffening test. This proves that the nonlinear behavior is not a result of complex nonlinear material representation, but it is mainly a result of the variable topology of the reinforced concrete element with a decreasing stiffness at every stage of cracking. This is clearly demonstrated in the model analysis and through several comparisons with test results.

The model simplicity allows us to gain valuable insight into the RC element behavior that cannot be obtained a priori by any numerical analysis where they are obtained only in the course of the ongoing incremental analysis.

As the reliability of the model is subjected to the limited slip range, it is mandatory to examine the slip range and verify that it falls within the specified limits. These limits were examined and found to generally enable the tension-stiffening analysis especially at the lower cracking levels. In cases where larger slips are developed, one should use a more complex bond–slip representation, although it will be less transparent and will not provide the insights that are gained with the present model.

**Author Contributions:** Conceptualization, D.Z.Y.; Formal analysis, Y.S.K. and V.R.F.; Investigation, D.Z.Y., Y.S.K. and V.R.F.; Methodology, D.Z.Y.; Supervision, D.Z.Y.; Writing—original draft, D.Z.Y.; Writing—review & editing, Y.S.K. and V.R.F. All authors have read and agreed to the published version of the manuscript.

**Funding:** This research received no external funding.

**Institutional Review Board Statement:** Not applicable.

**Informed Consent Statement:** Not applicable.

**Data Availability Statement:** The study did not report any data.

**Acknowledgments:** This work was supported by a joint grant from the Centre for Absorption in Science of the Ministry of Immigrant Absorption, the Committee for Planning and Budgeting of the Council for Higher Education under the framework of the KAMEA Program. The authors extend thanks to A. Abitbul for his help on this study.

**Conflicts of Interest:** The authors declare no conflict of interest.

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
