# Peer review of "Analytical Modeling of Crack Widths and Cracking Loads in Structural RC Members"

_infrastructures, doi:10.3390/infrastructures7030040_

Round 1

Reviewer 1 Report

Analytical modelling of crack widths and cracking loads in structural RC members is an interesting methodological article, proposed for a one-dimensional reinforced concrete element. I would suggest the following corrections before the publication:

- Line 57-65: Please add some references.

- Section »1.3 Tension-stiffening« gives very illustrative description of the research problem, but many statements are not supported by any reference. Please complete descriptions with appropriate references.

- Please unify the formatting of the image subtitle tags and formatting of diagram axis (the same font, font size, style, …)

- Line 448: Please add units to the given range of slope A.

- Table 1 and Table 2: Please add units.

- Line 360: Why is the same equation presented two times (duplication of the information)?

Reviewer 2 Report

The authors investigate the crack development in reinforced concrete structures. Analytical closed-form solutions expressions are provided for the number of expected cracks and their width. Comparisons with numerical and experimental data indicate a good agreement between the proposed formulas and existing literature results. The work is overall well-structured and of interest in crack response of RC structural members. There are certain issues that need to be however addressed. In particular:

  1. The proposed model is characterized as linear. However, both its formulation as well as the form of the responses obtained (Figs. 8-12) clearly demonstrate that such a naming is not representative. The term linear needs to be substituted by a descriptive modeling definition.
  2. The novelty of the work with respect to existing literature contributions needs to be clearly highlighted in the introduction section of the manuscript.
  3. The limitations of the model applicability with respect to the geometries (slenderness ratios) and boundary conditions need to be clearly defined.
  4. The order and positioning of the results presentation needs to be revised, while results need to be clarified: Fig. 8 appears to contain new results based on the model provided. However, the parameter setting for the results provided appears is rather poorly defined or merely referenced, while the rationale for not including the corresponding information in the results section 4 is rather unclear.
  5. The manuscript needs to be thoroughly proof-read and different typos and grammatical errors at various parts need to be corrected Indicatively: the title of section 3.3 needs to be corrected.

Reviewer 3 Report

Dear Authors,
thank you for your interesting paper focused on the origin and development of cracks in RC members. My comments are:
- line 46 - "akong", what does it mean? Isn't that "along the rebar"?
- do not use "The" in the title of the chapters, "1.2 Cracking problem ...", "2. New Approach", etc. is sufficient. You used "1. Introduction", not "1. The Introduction".
- FIG. 1 - small font, difficult to read. Please increase the font size.
- line 261 - The - capital "T" - start of sentence.
- I recommend writing their units in formulas and variables, e.g. [m], [m2] ...,
- line 308, formula (13-b) - is it "G1-F1", or "G1 = -F1"?
- line 360 ​​- why is equation (18-a) given again the second time?
- FIG. 2 and FIG. 3 - Variation - capital "V", the start of a sentence,
- lines 350 and 355 - I recommend using the designations of equations (18-a) and (18-b),
- line 412, should be, Viwathanatepa et al [36],
- use "where" after formulas (not capital "W"),
- tab. 3, stage "2" - delta 2 = delta 1 = 0,
line 581, title of paragraph 4. - shouldn't these be "analytical methods"? Not "analysis methods".
- lines 610, 611, 616, 619 ... - literature [45] is Chan et al., you have not literature Houde and Mirza in references, use Houde and Mirza given/mentioned in [45]
- line 620 - it is chapter 5. (not 4.).
Best regards.
